



# Quantitative Error Analysis on Polarimetric Phased Array Radar Weather Measurements to Reveal Radar Performance and Configuration Potential

Junho Ho[1,2], Zhe Li[3], Guifu Zhang[1,2]

[1] School of Meteorology, University of Oklahoma, Norman, Oklahoma, USA
     [2] Advanced Radar Research Center, University of Oklahoma, Norman, Oklahoma, USA
     [3] MathWorks, Natick, Massachusetts, USA

*Correspondence to*: Junho Ho (jhho0626@ou.edu)



**Abstract.** The initial weather measurements from two polarimetric phased array radars (PPAR) with cylindrical and planar configurations, both developed by the Advanced Radar Research Center (ARRC) at the University of Oklahoma (OU), were compared with those from the dish-antenna systems, the operational KTLX Weather Surveillance Radar-1988 Doppler (WSR-88D) located in Oklahoma City, Oklahoma (~23 km northeast of OU). Both the cylindrical PPAR (CPPAR) and the planar PPAR (PPPAR) in Horus are S-band two-dimensional (2D) electronic scan PPAR. This

comparison investigates the error statistics of the polarimetric measurements in one-dimensional (1D) electronic scan from each radar during two convective rain events. The first event occurred on 30 August 2019, when the CPPAR performed a 3.3° elevation plan-position indicator (PPI) scan at 25 azimuth angles. The second event took place on 4 October 2023, when Horus conducted range-height indicator (RHI) scans at 64 elevations. For both events, KTLX provided volumetric polarimetric radar data and served as the reference. To ensure temporal and spatial alignment

between the radars, reconstructed RHI scans and PPI sectors from KTLX were matched to the corresponding Horus rays and CPPAR domain, respectively. The standard deviations and mean biases of the PPAR weather measurements were calculated and analyzed. The standard deviations of the two PPARs were similar and met the Radar Functional Requirements set by the National Oceanic and Atmospheric Administration/National Weather Service. However, as noted in previous studies, the standard deviation, and biases of polarimetric variables from Horus exhibited varying

error characteristics depending on the electronic steering angle from broadside. The present results suggest that PPPARs may have difficulties in producing high-quality polarimetric data at large steering angles and further investigation on both CPPAR and 2D PPPAR is required to find the optimal design for future weather applications.

Key words: Polarimetric phased array radar (PPAR), cylindrical PPAR (CPPAR), Horus, planar PPAR (PPPAR),
error quantification.



## 1. Introduction

Phased array radars (PARs) are an emerging technology in the meteorological community. They offer the advantage of providing rapid and timely information that greatly enhances the understanding of severe weather phenomena as they unfold (Kuster et al., 2016; Wilson et al., 2017). PARs are also versatile and can effectively serve multiple purposes (Weber et al. 2007; Zrnic et al., 2007; Zhang and Doviak, 2007; Heinselman et al., 2008; Stailey and Hondl, 2016; Kollias et al., 2022). Many countries are actively involved in the development of PAR systems to replace or complement existing parabolic dish antenna operational radars (e.g., Wu et al. 2018, Kikuchi et al. 2020; Kollias et al., 2022; Palmer et al., 2022). Among PAR designs, the most common are 1D planar PARs, which have been investigated in the X-band polarimetric PAR (PPAR) in Japan (Kikuchi et al., 2020; Ushio et al., 2022), China (Wu et al., 2018; Yu et al., 2020), and the United States of America (USA, Wurman and Randall, 2001; Bluestein et al., 2010; Orzel and Frasier, 2018). 1D planar PPARs (PPPARs)—those with electronic scanning in elevation and mechanical steering in azimuth—can provide high-quality polarimetric data and represent a compromise between a costly-but-fully-electronic multi-face PPAR system and the less costly-but-slower-traditional rotating dish system. Preliminary error analysis and meteorological applications to improve weather forecasting using these 1D PPPARs have been performed (e.g., Orzel and Frasier, 2018; Kim et al., 2021; Baron et al., 2023).

In recent years, the most flexible and useful design of 2D PPARs for meteorological applications remains a subject of ongoing discussion since the formulation presented in Zhang et al., 2009. This is primarily due to the complexity and difficulty of providing high-quality polarimetric measurements when the beam steers off the broadside. Two main design approaches—the cylindrical design (Fulton et al., 2017; Golbon-Haghighi et al., 2021; Zhang, 2022; Zhang et al., 2011) and planar configurations (Heberling and Frasier, 2021; Palmer et al., 2022, 2023)—have garnered the most attention for 2D electronic scanning, each with its own advantages and disadvantages.

An S-band fully digital PPPAR named Horus (Fig. 1a) was developed by the Advanced Radar Research Center (ARRC) at the University of Oklahoma (OU) with funding from the National Severe Storms Laboratory (NSSL) and the Office of Naval Research (ONR) (Palmer et al., 2023). The fully digital design, with element-level analog to digital converters (ADC), can provide advantageous characteristics in multi-functionality, including high flexibility in spatio-temporal resolution and sampling, beam agility, interference mitigation, and, in theory, software configurability. However, as a 2D PPPAR, Horus faces major challenges in calibrating polarimetric variables to meet weather observation requirements ( Zhang et al., 2009, 2011; Lei et al., 2013, 2015; Palmer et al., 2023). Fundamental issues affecting data quality include geometrically induced copolar biases, cross-polarization coupling and sensitivity loss as well as performance degradation as the beam steers off broadside (Zhang et al., 2011; Zhang 2016; Zrnic et al., 2011). PPPARs utilize hundreds of beams with different characteristics, which necessitates beam steering-dependent calibration (Ivić et al., 2019; Weber et al., 2021). The most critical issue is the sensitivity loss and performance degradation when steering at wide angles off broadside; while the bias can be corrected, addressing the sensitivity loss is difficult and may require increased antenna size and higher transmit power to meet the sensitivity requirements at large off-broadside angles (Zhang et al., 2011). Also, the polarization purity loss is difficult to compensate/calibrate, although calibration methods (e.g., Zhang et al. 2009, Dorsey et al. 2021, Fulton et al., 2022; Ivić, 2023) to mitigate



cross-polar biases have been proposed. High quality polarimetric weather measurements have not been achieved with error quantification yet when a PPPAR steers at wide angles.

Alternatively, the cylindrical PPAR (CPPAR) design has been proposed and demonstrated, and a prototype was developed by the ARRC (Fig. 1b) due to its advantageous properties of scan-invariant azimuthal beams and polarization orthogonality in all directions (Zhang et al., 2011; Karimkashi and Zhang, 2013, 2015; Fulton et al., 2017; Golbon-Haghighi et al., 2021). CPPAR provides more efficient radiating power and spectrum utilization without the need for face-to-face matching. These features make CPPAR capable of delivering effective and efficient polarimetric weather observations compared to the planar design (Zhang et al., 2011; Li et al., 2021; Dorsey et al., 2022; Logan et al., 2022). Nevertheless, several challenges of CPPAR have also been mentioned such as its relatively new design and development, the all-in-one system, whereas the PPPAR would operate 4 faces independently for different directions, and the potential influence of creeping waves and interferences (NSSL, 2014). These challenges have been studied and addressed through design/development of high performance radiating elements and optimized beamforming with active element patterns, in which the creeping wave effects have been taken into account (Golbon-Haghighi et al., 2021; Li et al., 2021; Mirmozafari et al., 2019; Zhang, 2022).

As mentioned above, many of the properties and characteristics of the two 2D PPAR systems have been explored based on the physical understandings of the electromagnetic (EM) theory, simulations, and experiments. In addition, the hardware requirements and specific calibration procedures for Horus and CPPAR have been discussed in previous studies (e.g., Li et al., 2021; Palmer et al., 2023). The primary objective of this study is to compare the error statistics of weather observations collected by CPPAR and Horus. This comparison aims to assess the quality of the polarimetric data in their current states, investigate the potential issues, and clarify any misunderstandings about the two configurations. This study will incorporate findings from previous research conducted over the last 10 years on 2D PPAR development. It should be noted that the two radars are at different stages of development and with different levels of investment, and the weather observations are not from the same weather event. However, the comparison results presented in this study represent the first observation-based comparison of the two radar configurations. This result will be valuable in guiding the selection of the optimal configuration of PPARs for meteorological applications.

The current specifications of CPPAR and Horus, along with the reference measurement—KTLX, a nearby operational WSR–88D radar (Fig.1c)—are presented in Section 2. Section 3 illustrates calculations of the standard deviations of the Horus and CPPAR observations, and comparisons between the Horus and KTLX data, as well as between the CPPAR and KTLX data, with the mean bias and related statistics quantified. In Section 4, the off-broadside impact of planar design is explored and discussed. Section 5 discusses advantages and disadvantages of CPPAR and 2D PPPAR configurations as well as their potential. Finally, Section 6 summarizes the results and discusses possible development directions and improvements of the PPARs for weather measurements.



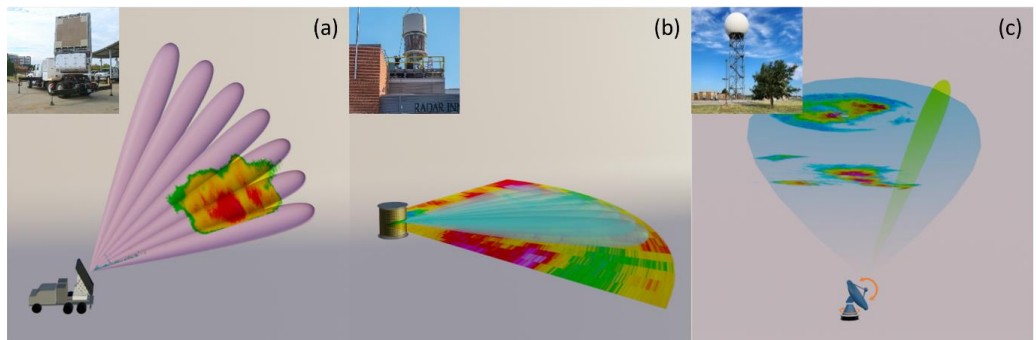

100    **Figure 1.** Depictions of (a) Horus, (b) CPPAR, and (c) WSR–88D (KOUN) scanning strategies based on the current configuration for weather measurements. The Horus image is taken from Palmer et al. (2023).

## 2. Data and methods

### 2.1 Horus and CPPAR experiment configuration

The Horus radar system has a planar design with $5 \times 5$ panels, each panel consisting of $8 \times 8$ dual-polarization
105    antenna elements. Its full aperture size is 2.03 m $\times$ 2.03 m, and it operates at the S band at approximately 3.07 GHz (Table 1), as documented in Palmer et al. (2023). For this study, two sets of measurements are examined when the radar was configured with 13 out of 25 panels, with a transmit power of 8.32 kW and an antenna gain of approximately 31.5 dB. The half-power beamwidth is 3.3° in both azimuth and elevation. Currently, only range-height indicator (RHI) scans have been performed with 64 elevation angles at 1° intervals (Figure 1a). A total of eight cases have been
110    measured by Horus, with six cases occurring prior to August 2023 using 5 panels and two subsequent cases using 13 panels. The bandwidth was approximately 7.8 MHz resulting in a range resolution of 19.2 m, with pulse compression. The progressive pulse compression technique (Salazar Aquino et al., 2021) was utilized to remove the blind range, which used to be about 4.8 km. The temporal resolution was approximately 4 seconds with a pulse repetition time (PRT) of 1 ms with 64 samples per dwell. The scanning strategy consists of a mechanical inclination of 31.5° with
115    scans ranging from −31.5° to 31.5° in elevation (i.e., ~0° to 63° ground-relative elevation angles) at 1° intervals.

| Radar Parameters | Horus | CPPAR | KTLX |
|---|---|---|---|
| Frequency (GHz) | 3.07 | 2.76 | 2.8 |
| Transmit power (kW/polarization) | 8.32 | 4.32 | 375 |
| Antenna gain (dB) | 31.5 | 26 | 45.5 |
| Elevation beamwidth (°) | 3.3 | 5.35 | 0.925 |
| Azimuth beamwidth (°) | 3.3 | 6.2 | 0.925 |

**Table 1.** Specifications relevant to the sensitivity of each PAR configuration.

The 2-m prototype of the CPPAR, installed on the roof of the ARRC, also operates at the S band at a frequency
120    of 2.76 GHz (Table 1). An illustration of the CPPAR is shown in Figure 1b. CPPAR allows a single beam for mechanical scans and 25 commutating beams for electronic scans, with a PRT of 1 ms for 64 pulses per dwell. The range sampling interval is approximately 30 m, with the first 170 gates representing a blind range of about 5.1 km.



The CPPAR consists of a total of 96 subarrays/columns (although only half of them, 48 columns, are active due to the budget constraint), each with an azimuthal spacing of 3.75° consisting of 19-element linear arrays. Of these 48 columns, 24 columns are used to form an electronic beam, yielding a total of 25 beams for the electronic scan, and among them the central beam sector (No. 13−36 columns) is used for the mechanical scan in the study. The azimuthal beamwidth is approximately 6.2° after tapering, and the elevation beamwidth is 5.35° (Table 1). The transmit/receive antenna gain is 26 dB, and the peak transmit power is 4.32 kW (Figure 1b). However, the development of CPPAR has been halted to its current state, and only single time step plan-position indicator (PPI) scans had been performed. Further specifications of CPPAR can be found in Li et al., (2021).

The radar parameters of the two PPARs listed in Table 1 exhibit large differences in transmit power, beamwidth, and antenna gain compared to KTLX (Figs. 1c vs. 1a and 1b). The sensitivity difference derived from these parameters can be further discerned from the minimum detectable reflectivity plot (Fig. 2). The plot illustrates the calculated values from the lag-0 estimates of weather measurements (solid) from Horus (red), CPPAR (blue), and KTLX (black), compared to those from their radar parameters (dashed), as a function of range. Note that the system calibration factor of each radar was slightly adjusted to better align the two lines. As expected from the parameters of the three radars, the two 2D PPARs have much lower sensitivity of 25 dBZ for CPPAR, and 10 dBZ for Horus, compared to −10 dBZ for the operational KTLX at 45 km away from the radar. Even between the two PPARs, the difference is considerable, with Horus ~ 15 dB better than CPPAR.

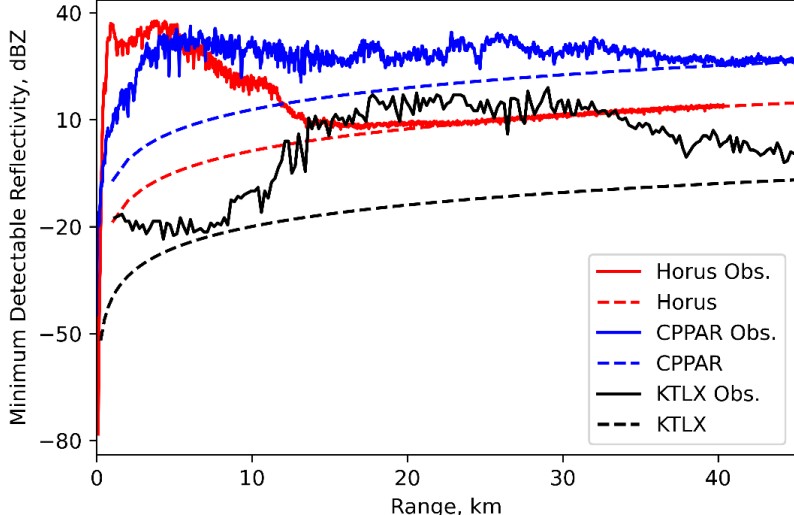

**Figure 2.** Plot of the minimum detectable reflectivity of Horus (red lines), CPPAR (blue lines), and KTLX (black lines). The solid lines are estimated from the weather measurements, and the dashed lines radar parameters.

### 2.2 Reference measurements

The three radar variables ($Z_H$, $Z_{DR}$, $\rho_{hv}$) from the operational KTLX radar located at Oklahoma City (34.33°N, 97.21°W) were used as a reference to calculate the mean biases and standard deviation of the differences between the

radar measurements. Figure 3 depicts the relative positions of Horus (3a) and CPPAR (3b), represented by blue dots, respectively, in relation to KTLX (black dot), as well as the direction/sector of interest on the KTLX PPI for each case. The KTLX beams do not exactly coincide with the Horus RHI and the CPPAR PPI scans because the systems are not co-located and the beamwidths of the two radars are very different; therefore, other variables ($v_r$, $\sigma_v$, $\Phi_{DP}$) were

excluded from the comparison due to their radial dependency.

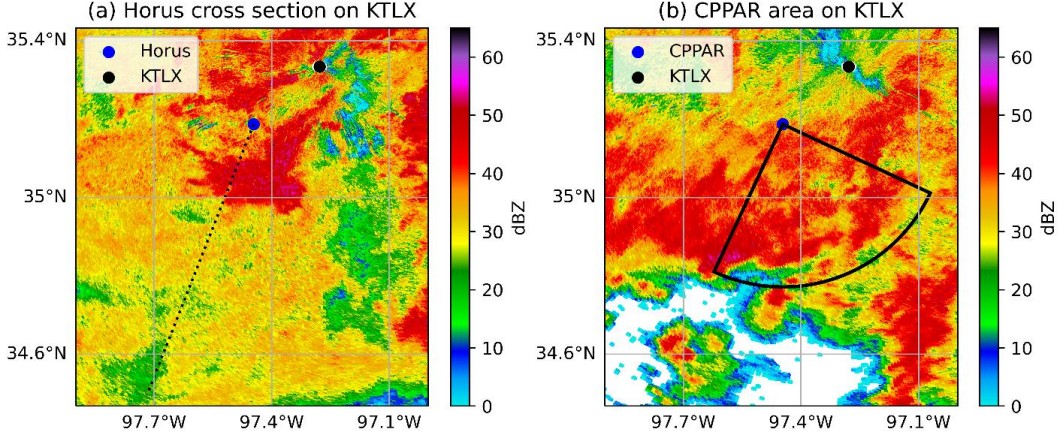

**Figure 3.** Plot of the observed reflectivity ($Z_H$) from KTLX for regions of the Horus (left) at 2239 UTC on 4 October 2023, and CPPAR (right) at 1505 UTC 30 August 2019. The blue dot denotes Horus and CPPAR, and the black dot KTLX. The dashed line represents the azimuth of the Horus RHI scan, and the black solid lines the edges of the CPPAR PPI scan.


     In order to minimize the influence in the difference in the positions of the radars, each elevation and time was carefully matched by selecting the Horus rays from the best-matching KTLX observation time for each elevation angle. The time can be well matched in the case of Horus and KTLX because Horus provides 4-second updates. In addition, three different types of interpolation grids were used to fit the KTLX reconstructed RHI to the Horus observations: (1)

both KTLX and Horus were interpolated to the same grid with grid spacings of 10 m horizontally and 125 m vertically; (2) KTLX data were interpolated to the Horus range and elevation angles; and (3) both Horus and KTLX were objectively analyzed to a common grid of 500 m horizontally and 125 m vertically, respectively. Note that nearest-neighbor interpolation was used for both the KTLX reconstructed RHI and the gridded Horus data. Although not shown in subsequent figures, all interpolated grids showed similar results and had little effect on the error statistics.

Therefore, the KTLX reconstructed RHI were converted to match the Horus RHI scans for easier error quantification in the subsequent analyses.

     For CPPAR, it is not possible to match the beams perfectly in elevation and time because only one PPI data were collected for the case. The beam height of the 3.3° scan from the roof of the RIL building to approximately 45 km in range is about 2.8 km AGL, the closest elevation from the KTLX data was used for the same spatial coverage of

CPPAR. Similar to the matching method with Horus, the KTLX data were extracted by interpolating the nearest points to the cross section between the CPPAR and the calculated location of 40 km in range for each azimuth angle. It

should be noted that, even beyond the aforementioned matching in time and space, inherent mismatches persist between the interpolated RHIs or PPIs due to differences in radar resolution, beam width, and location.

### 2.3 Error statistics calculations

The standard deviation calculation for polarimetric radar measurements has traditionally been performed using spatial sampling, assuming a locally homogeneous precipitation field. This typically employs samples from a number of range gates, as demonstrated in an earlier CPPAR data analysis using $N=17$ range gates (Li et al., 2021) and shown in Equation (1),

$$STD = \sqrt{\frac{1}{2(N-m)} \sum_{l=n-(N-1)/2}^{n+(N-1)/2-m} (X_{l+m} - X_l)^2} \qquad (1)$$

where $n$ is the gate number at which the standard deviation is estimated, and $X_l$ is the polarimetric data at gate $l$, and $X_{l+m}$ is the value at $l+m$ gates, with $m$ as the spatial step after employing spatial samples. Various values of $m$ have been tested for spatial samples, with a slight increase in standard deviations as m increases. The value $m = 2$ has been selected to avoid potential overlap in range samples, Considering the small sample spacing, the local homogeneity should be a valid assumption. To maintain statistical significance, only gates with no missing data (i.e., all 15 samples

from 17 range gates) were used to calculate the standard deviation.

In addition to spatial sampling, given the assumptions of ergodicity and local stationarity that apply to the Horus data due to its rapid updates every 4 seconds, it is possible to compute the standard deviation of the radar data from temporal samples. This approach involves examining the differences between successive time steps (i.e., $m = 1$) over the entire dataset. Since different range gates observe distinct parts of the precipitation field, and the movement of storms within 4 seconds generally falls within the resolution volume, but the signals are decorrelated, using temporal

samples can provide more accurate estimates in many cases without the assumption of spatial homogeneity. The standard deviation is calculated for various polarimetric variables, including $Z_H$, velocity ($v_r$), spectrum width ($\sigma_v$), $Z_{DR}$, $\rho_{hv}$, and differential phase shift ($\Phi_{DP}$), for both spatial and temporal sampling. In the spatial sampling approach, 17 range gates were used based on the middle time step to calculate the standard deviation. Experimenting with

different time steps or increasing the number of samples did not significantly change or improve the standard deviations. The computed values for both spatial and temporal samples are compared to theoretical values and the radar functional requirements set by the National Oceanic and Atmospheric Administration/National Weather Service (NOAA/NWS RFR). The theoretical values were derived using lag-0 estimate equations from Doviak and Zrnic (2006).

### 3. Comparison and validation of Horus and CPPAR data

To assess the data quality and system performance of the weather measurements, the error characteristics of the polarimetric data are calculated and quantified. Horus started its first weather observations in December 2022 and continues to observe cases of shallow and deep convective precipitation. This study specifically examines a recent convective precipitation event on 4 October 2023 from 22:19 to 22:45 UTC, focusing mainly between 22:36 to 22:45



UTC. As illustrated in Figure 3a, the Horus beam was directed at an azimuth of 198° from the north, penetrating the convective region of the storm. The 18:00 UTC Norman sounding of 04 October 2023 reveals favorable environmental conditions for deep convective storms, with a convective available potential energy (CAPE) of ~3078 J kg$^{-1}$. The combination of abundant low-level moisture and diurnal boundary layer heating with an approaching mid-level shortwave trough provided favorable conditions for thunderstorm development. The group of isolated convective cells

of interest originated near the western Oklahoma/northwestern Texas border around 17:50 UTC, and a band of supercells in a loosely organized mesoscale convective system (MCS) moved across Oklahoma. According to the NWS, several reports of strong winds and hail were documented throughout central and northeastern Oklahoma (https://www.spc.noaa.gov/exper/archive/event.php?date=20231004).

    Until the summer of 2020, CPPAR underwent development/testing and conducted weather measurements. This

study focuses on weather observations that took place on 30 August 2019 at 15:04 UTC. Figure 3b depicts the measurement area of the CPPAR as observed from KTLX at 19:14 UTC. Like the convective case for Horus, the atmospheric conditions were conducive to severe storms. Around 10 UTC, the preexisting MCS from Kansas continued to move southeastward to produce thunderstorms in central Oklahoma. CAPE of up to 4000 J kg$^{-1}$ and steep lapse rates of 2-6 °C km$^{-1}$ have been reported, maintaining moderate instability ahead of the MCS. The NWS recorded

strong       gusts       of       up       to       71       mph       in       northern       to       central       Oklahoma (https://www.spc.noaa.gov/exper/archive/event.php?date=20190830).

    In addition to the storm events, some radar parameters, waveforms, and calibration techniques differ between the two 2D PPARs. For example, Horus employed progressive pulse compression (Salazar Aquino et al., 2021) to eliminate the blind range and used mutual-coupling based calibration (e.g., Fulton et al. 2022; Palmer et al. 2023). On

the other hand, CPPAR used a simpler calibration method by mounting a calibration horn on top of a nearby building to optimize the beams. This method aimed to match copolar patterns, maximize gain, and minimize sidelobe levels and cross-polar biases (Li et al., 2021). Also, a pulse compression waveform was used with a pulse width of 34 μs, resulting in short-range blind range of approximately 5.1 km for CPPAR. The timeseries data from the two PPARs were processed in the same way using lag-0 estimates for regions with SNR greater than 20 dB and lag-1 for the rest.

It should be noted that both the software and hardware of KTLX have not undergone any significant changes between 2019 and 2023 (https://training.weather.gov/wdtd/buildTraining/RPG-RDA.php).

    The initial comparison is conducted between the radar measurements from the operational radar, KTLX, and the two PPARs to compute the mean bias and the standard deviation of the differences. Subsequently, a comparative analysis of the standard deviation is derived from both spatial and temporal samples for Horus, and solely spatial

sampling for CPPAR.

### 3.1 Bias calculation of Horus and CPPAR data

    The spatial distribution of the polarimetric variables from the two radars shown in Figure 4 provides valuable information for identifying potential system deficiencies and understanding the error characteristics of Horus. KTLX, which benefits from higher antenna gain and transmit power, exhibits significantly higher SNR and sensitivity

compared to Horus (Figs.4a vs. 4b). The convective core located between 10 and 20 km demonstrates good agreement





between the radars (Figs.4c vs. 4d). However, there are still some clear differences in the magnitude of the measurements. For example, the maximum $Z_H$ near the ground, at about 15 km, exhibits a difference of more than 5 dB, and a lack of sensitivity in the Horus data remains apparent at further distances. Despite the bias, the ability of Horus to capture true RHIs provides much more detailed microphysical and dynamical process information due to the
improved spatial and temporal resolution, demonstrating the potential of PARs to improve meteorological applications.

The polarimetric variables from Horus show more notable differences with that from KTLX. Slight $Z_{DR}$ bias exists, with ~1.0 dB difference near the convective core (Figs. 4e vs. 4f). Additionally, the low $Z_{DR}$ region between ~25 and 30 km for KTLX does not appear clearly for Horus, and noisy values of up to 1 dB above the melting layer. Overall, the $Z_{DR}$ values from Horus agree well with KTLX, with positive biases of less than 0.5 dB throughout the
entire domain. Note that KTLX has limited observations in the lower elevations due to its distance from the storm, and the near-ground data are interpolated from higher altitudes, leading to relatively larger differences in the lowest elevations. In addition, since the error characteristics of the polarimetric variables depend on $\rho_{hv}$, the lower to middle elevations of $Z_{DR}$ may also be affected by the degraded $\rho_{hv}$ in these regions (Fig. 4g). The high $\rho_{hv}$ in Horus is notable in the lower and upper elevations, and the melting level agrees well with KTLX (Figs. 4g vs. 4h). However, Horus
shows reduced $\rho_{hv}$ values of less than 0.94 in the mid-altitude regions from 1.5 to 6 km, and these are more pronounced from about 20 km. While the reduction can be partly explained by snow melting, low SNR, and propagation effect, the reduced $\rho_{hv}$ along the entire radials is a concerning feature. This $\rho_{hv}$ reduction can imply error associated with electronic steering at large angles away from the broadside. Future improvements in PPAR signal processing for weather applications are planned in light of the observed biases in the $Z_{DR}$ and $\rho_{hv}$, and the need to minimize the
influence of clutter and contamination by addressing sidelobes, beam width and steering loss issues.

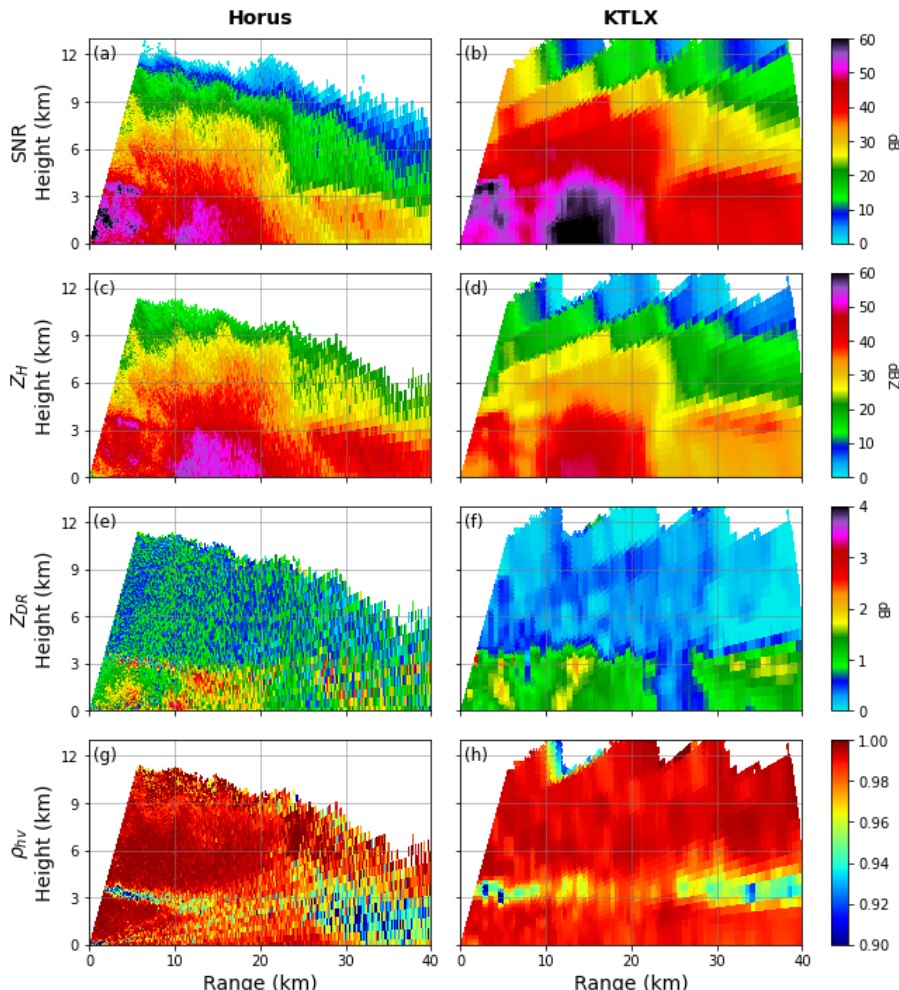

**Figure 4.** Spatial distribution of SNR, $Z_H$, $Z_{DR}$, and $\rho_{hv}$ from Horus (a, c, e, and g) and KTLX (b, d, f, and h) measurements on 4 October 2023. All variables for both radars are plotted for SNR greater than 10 dB.

The mean bias and standard deviation of the differences from the KTLX comparison for the case shown in Fig.4 are organized in Table 2. The values remain relatively stable and consistent across different SNR ranges. The mean bias and standard deviations for $Z_H$ remains around 3 to 4 dB and 5 to 6 dB, respectively (Tables 2a and b). $Z_{DR}$ shows slight improvements with SNR decreasing to 0.23 dB bias and 0.72 dB standard deviation. The standard deviation values are typically similar to or greater than the corresponding mean bias values, with limited influence of beam

broadening or mismatch on these statistics. There is minimal improvement in the accuracy of $\rho_{hv}$ measurements with increasing SNR, suggesting that such a reduction is not due to lower SNR values. The consistently low $\rho_{hv}$ bias is of concern, especially for the significant $\rho_{hv}$ reduction between 20 and 40 km shown in Figure 4. Further analysis to investigate the potential cause of this degradation is discussed in section 4.





(a) Mean bias

|  | SNR $\geq$ 0 | SNR $\geq$ 5 | SNR $\geq$ 10 | SNR $\geq$ 15 | SNR $\geq$ 20 |
|---|---|---|---|---|---|
| $Z_H$ (dB) | 3.06 (3.63) | 3.27 (3.76) | 3.49 (3.94) | 3.72 (4.15) | 3.99 (4.33) |
| $Z_{DR}$ (dB) | 0.27 (0.32) | 0.26 (0.32) | 0.25 (0.31) | 0.24 (0.3) | 0.23 (0.29) |
| $\rho_{hv}$ | −0.004 (0.002) | −0.003 (0.003) | −0.002 (0.003) | −0.002 (0.003) | −0.002 (0.003) |

(b) Standard deviation

|  | SNR $\geq$ 0 | SNR $\geq$ 5 | SNR $\geq$ 10 | SNR $\geq$ 15 | SNR $\geq$ 20 |
|---|---|---|---|---|---|
| $Z_H$ (dB) | 5.96 | 5.79 | 5.56 | 5.33 | 5.05 |
| $Z_{DR}$ (dB) | 0.81 | 0.77 | 0.75 | 0.74 | 0.72 |
| $\rho_{hv}$ | 0.036 | 0.032 | 0.032 | 0.032 | 0.033 |

**Table 2.** Mean bias and standard deviation of the differences between Horus and KTLX for each SNR range. The values in parentheses denote the median values.


The comparisons between CPPAR and KTLX are depicted in Figure 5, which show a generally similar spatial distribution as that between Horus and KTLX. The magnitude and distribution of $Z_H$ from CPPAR agrees well with those from KTLX, except for a region of low SNR in the far ranges and a few radials influenced by a water tower close to the radar (Figs.5 a vs. 5b). Note that the CPPAR SNR is much lower than that of Horus and KTLX, as expected

from the radar parameters. In the far ranges, CPPAR data are absent or have much smaller values, while KTLX shows values close to 20 dBZ (Figs.5c vs. 5d). This discrepancy can be attributed to the distinct transmit power and sensitivity of the two systems. The $Z_{DR}$ comparisons display similar trends, with a common location of $Z_{DR}$ values up to 3.5 dB at around 40 km in range (Figure 5e vs. 5f). $Z_{DR}$ from CPPAR closely align with those from KTLX, showing no notable bias. In the lower SNR region, large $\rho_{hv}$ values appear at the far edge of the CPPAR measurements, with

reduced $\rho_{hv}$ along radials with low SNR and/or the presence of the water tower in directions at azimuth of ~133º from the north (Figs. 5g vs. 5h). It is important to note that some differences in elevation and time between CPPAR and KTLX are unavoidable given that KTLX scans only every 6 min, and CPPAR did not perform additional scans at different times. There has been a misconception that cylindrical configuration may be prone to interferences or creeping waves (e.g., NSSL, 2014). However, as demonstrated from previous studies based on physical formulations

(e.g., Golbon-Haghighi et al., 2021 and Li et al. 2021), such creeping wave effects are not noticeable in the CPPAR measurements. These results highlight the potential differences and biases between the two systems and provide valuable insights into the performance and error characteristics of the emerging CPPAR in comparison to the more established KTLX radar.



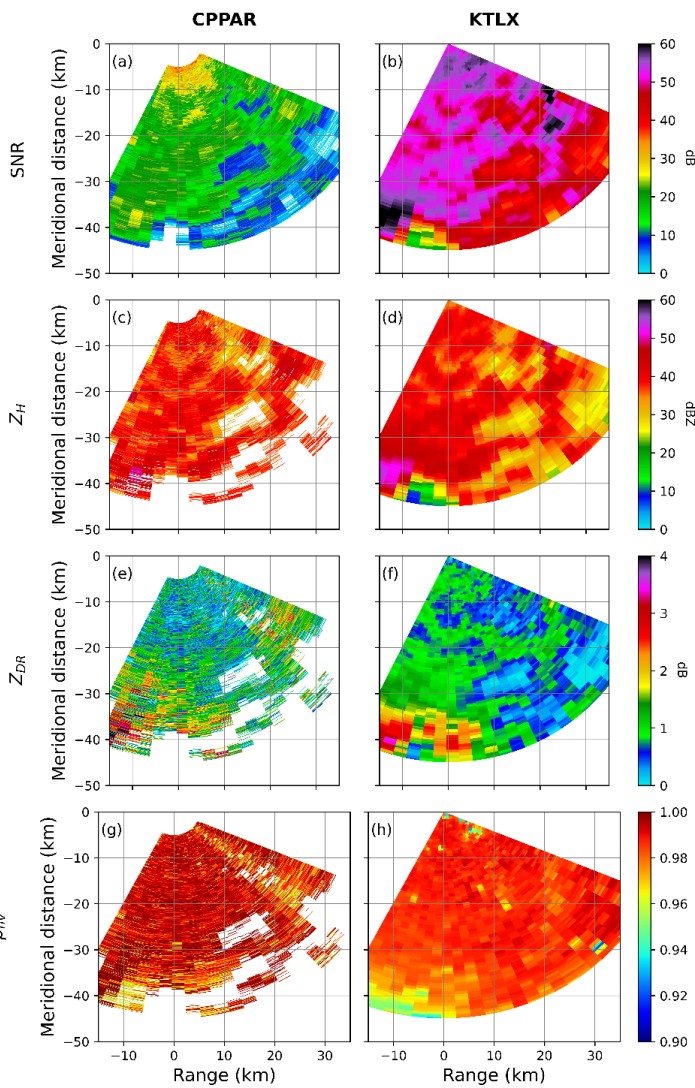

**Figure 5.** Same as Figure 4, except for CPPAR and KTLX.

The calculated mean bias and the standard deviation of the differences between CPPAR and KTLX are organized in Table 3. The magnitude of the mean bias in all radar variables remains relatively stable with SNR $\geq$ 20 dB of approximately $-1.29$ dB, 0.04 dB, and 0.009 for $Z_H$, $Z_{DR}$, and $\rho_{hv}$, respectively. The small polarimetric biases compared to that of Horus is a promising feature. Especially, the consistently low and positive bias of $\rho_{hv}$ is an encouraging feature and implies that CPPAR is making progress towards achieving more accurate and reliable weather measurements (Table 3a). The standard deviation of the differences for CPPAR is generally similar to that of Horus for $Z_H$, $Z_{DR}$, and $\rho_{hv}$ (Table 3b). The relatively larger overall biases in the Horus data may be attributed to errors in polarimetric calibration methods used as well as inherent design issues.


(a) Mean bias

|  | SNR ≥ 0 | SNR ≥ 5 | SNR ≥ 10 | SNR ≥ 15 | SNR ≥ 20 |
|---|---|---|---|---|---|
| $Z_{\mathrm{H}}$ (dB) | −1.29 (−1.26) | −1.07 (−1.13) | −0.82 (−0.98) | −0.63 (−0.81) | −1.29 (−1.26) |
| $Z_{\mathrm{DR}}$ (dB) | 0.04 (0.00) | 0.03 (0.00) | 0.03 (0.00) | 0.01 (−0.02) | 0.04 (0.00) |
| $\rho_{\mathrm{hv}}$ | 0.009 (0.007) | 0.006 (0.006) | 0.006 (0.006) | 0.007 (0.006) | 0.009 (0.007) |

(b) Standard deviation

|  | SNR ≥ 0 | SNR ≥ 5 | SNR ≥ 10 | SNR ≥ 15 | SNR ≥ 20 |
|---|---|---|---|---|---|
| $Z_{\mathrm{H}}$ (dB) | 5.22 | 5.05 | 4.83 | 4.59 | 4.17 |
| $Z_{\mathrm{DR}}$ (dB) | 0.96 | 0.85 | 0.77 | 0.73 | 0.69 |
| $\rho_{\mathrm{hv}}$ | 0.065 | 0.053 | 0.052 | 0.054 | 0.04 |

**Table 3.** Same as Table 2, except for difference between CPPAR and KTLX.

### 3.2 Standard deviation estimates of Horus and CPPAR data

Based on the Horus and CPPAR measurements (Figure 4a, 4c, 4e, and 4g and Figure 5a, 5c, 5e, and 5g), the standard deviation of the two measurements using spatial and temporal samples are depicted in Figure 6 and Figure **7** to further investigate the error characteristics. Note that the standard deviation of the differences between the PPARs and KTLX in the previous section are different from the standard deviation estimates of the PPARs' measurements in this section. There are differences between the two PPARs; Horus has a beamwidth of 3.3°, but the elevation angles

are sampled at 1° intervals. On the other hand, CPPAR used only 25 beams for the 90° sector with a beamwidth of 5.35°, which limits the influence of azimuthal oversampling. Therefore, the standard deviations were calculated along the ranges to avoid the influence of oversampling in the standard deviation estimates. In addition, the number of samples for Horus is ~3.6 times larger than CPPAR, and is even larger for higher SNR ranges.

          Figure 6 illustrates the standard deviation estimates of six radar variables based on spatial and temporal samples for SNR larger than 10 dB. The SNR plots are shown in Figs. 6a and 6b as a reference. While the magnitudes between

the spatial and temporal differ, the pattern of high standard deviation matches well (e.g., Figure 6c and 6e). For $Z_{\mathrm{H}}$, higher standard deviation values up to 2.5 dB are observed based on temporal samples, with ~1.5 to 2 dB throughout the majority of the RHI scan (Figure 6c and 6e). The standard deviation values are relatively consistent with few larger values in the lowest elevations due to ground clutter effects, melting layer, and in low SNR regions for both spatial

and temporal. Note that the very near ranges (i.e., < 5km) in the original blind range may have higher standard deviation values due to the influence of progressive pulse compression (Salazar Aquino et al., 2021). The $v_{\mathrm{r}}$ and $\sigma_{\mathrm{v}}$ also show similar features (Fig. 6g, 6i, 6k, 6m). The most concerning features, however, are the stripes of increased standard deviation of $Z_{\mathrm{DR}}$, $\rho_{\mathrm{hv}}$, and $\Phi_{\mathrm{DP}}$ observed at low to mid elevations, which may be related to the performance degradation as the electronic beam is steered away from the broadside, in addition to the physics of melting that can

affect multiple beams due to the relatively large beamwidth (Fig. 6d, 6f, 6h, 6j, 6l, 6n). These strips of high standard deviation are consistent with the reduced $\rho_{\mathrm{hv}}$ from the Horus measurements (Figure 4). The strips of increased standard deviation along the radials are also noticeable at higher elevation angles. These features, which are only noticeable



from the polarimetric variables, reinforce the possibility that these stripes are contributed by the inherent limitation of 2D PPARs when electronically scanning off broadside. Overall, the standard deviation values in Figure 6 exhibit reasonable distributions, consistent with the results in Table 4.

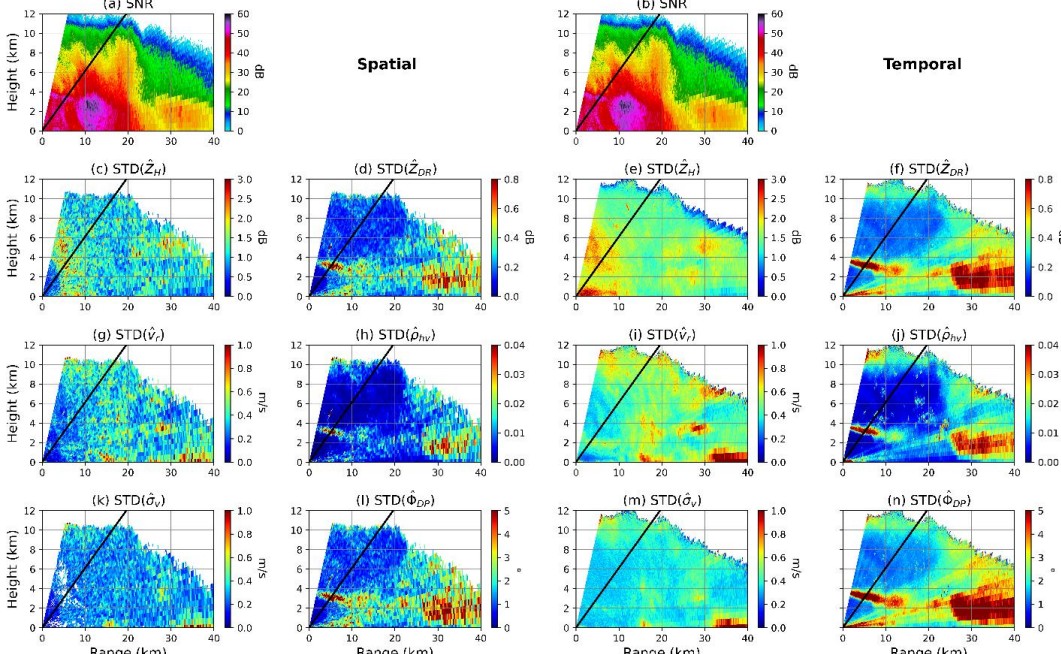

**Figure 6.** Spatial distribution of SNR, and standard deviations of reflectivity ($Z_H$), radial velocity ($v_r$), spectrum width ($\sigma_v$), differential reflectivity ($Z_{DR}$), correlation coefficient ($\rho_{hv}$), and differential phase shift ($\Phi_{DP}$) for SNR greater than 10 dB from Horus weather observations based on 146 timesteps on 4 October 2023. The first column uses 17 spatial gates, and the second column only temporal samples. The black lines denote the broadside direction.

The calculated values for both the spatial and temporal samples are compared with theoretical values and the radar functional requirements set by the National Oceanic and Atmospheric Administration/National Weather Service (NOAA/NWS RFR). The theoretical values are derived using equations from Doviak and Zrnic (2006). The values of $\sigma_v$ and $\rho_{hv}$, which influences the theoretical values of other variables, have been selected using the median value of the middle timestep for spatial samples, and all timesteps for temporal. The median values of calculated standard deviations for each signal-to-noise ratio (SNR) range, theoretical values, and the NOAA/NWS RFR are organized in Table 4, based on spatial (Table 4a) and temporal (Table 4b) sampling using Horus data. As expected, the standard deviation of the polarimetric variables generally exhibits a decreasing trend with increasing SNR. However, for larger SNR ranges, slight variations are expected due to the smaller number of data points available. For SNR values exceeding 20 dB, the standard deviations are even smaller than the theoretical calculations, especially for $v_r$ and $Z_{DR}$. In addition, there are considerable differences (~ 1.4 times larger for temporal) in the standard deviations between the spatial and temporal samples for all SNR regions. For instance, the standard deviations of $Z_H$ of the temporal samples within the SNR range of 5 to less than 20 dB are about 1.4 times larger for the spatial samples. This kind of differences is also true for $Z_{DR}$ and $\Phi_{DP}$. This could be attributed to the influence of various filters applied to the Horus data



(Palmer et al. 2023), including a clutter filter (Siggia and Passarelli, 2004) and a radio frequency interference filter (Cho, 2017). It emphasizes the need to account for these filters in subsequent analyses. Nevertheless, the standard deviation for SNR values greater than 20 dB agrees well with the theoretical calculations and are less than the NOAA/NWS RFR limits for the examined case.


(a) Spatial

| | 0≤SNR | 5≤SNR<10 | 10≤SNR<15 | 15≤SNR<20 | 20≤SNR | Theory | NOAA/NWS RFR |
|---|---|---|---|---|---|---|---|
| $Z_H$ (dB) | 1.10 | 0.85 | 0.91 | 0.86 | 1.09 | 1.39 | 1.8 |
| $v_r$ (m/s) | 0.31 | 0.47 | 0.40 | 0.40 | 0.29 | 0.43 | 1.0 |
| $\sigma_v$ (m/s) | 0.26 | 0.41 | 0.30 | 0.29 | 0.24 | 0.32 | 1.0 |
| $Z_{DR}$ (dB) | 0.20 | 0.47 | 0.31 | 0.19 | 0.18 | 0.32 | 0.3 |
| $\rho_{hv}$ | 0.005 | 0.032 | 0.015 | 0.010 | 0.003 | 0.005 | 0.006 |
| $\Phi_{DP}$ (°) | 1.40 | 3.24 | 2.17 | 1.42 | 1.20 | 2.07 | 2.0 |

(b) Temporal

| | 0≤SNR | 5≤SNR<10 | 10≤SNR<15 | 15≤SNR<20 | 20≤SNR | Theory | NOAA/NWS RFR |
|---|---|---|---|---|---|---|---|
| $Z_H$ (dB) | 1.65 | 1.02 | 1.04 | 1.01 | 1.69 | 1.49 | 1.8 |
| $v_r$ (m/s) | 0.47 | 0.67 | 0.53 | 0.51 | 0.44 | 0.45 | 1.0 |
| $\sigma_v$ (m/s) | 0.36 | 0.54 | 0.39 | 0.36 | 0.35 | 0.32 | 1.0 |
| $Z_{DR}$ (dB) | 0.33 | 0.60 | 0.38 | 0.27 | 0.25 | 0.33 | 0.3 |
| $\rho_{hv}$ | 0.010 | 0.038 | 0.018 | 0.011 | 0.005 | 0.005 | 0.006 |
| $\Phi_{DP}$ (°) | 2.29 | 4.13 | 2.69 | 1.96 | 1.72 | 2.23 | 2.0 |

**Table 4.** The median value for the standard deviation of six radar variables based on both (a) the spatial and (b) the temporal domain of the Horus data for five different signal-to-noise ratio (SNR) ranges. The theoretical values are selected based on middle timestep for spatial, and all 146 timesteps for temporal. The NOAA/NWS RFR represents the radar functional requirements of the NOAA and National Weather Service.


The standard deviation estimates for the CPPAR measurements are shown in Figure 7. As expected from the weaker power and lower sensitivity, wider beamwidth, less angular oversampling, and no spatial filtering, CPPAR exhibits larger standard deviation estimates, especially in the low SNR regions (Fig. 7a). $Z_H$ shows consistent values

of ~1.2 dB (Fig. 7b) except in the low SNR regions. For $v_r$ and $\sigma_v$, the values are ~0.6 m/s and 0.3 m/s, respectively (Figs. 7d and f). For $Z_{DR}$, $\rho_{hv}$, and $\Phi_{DP}$, the values fall below 0.3 dB (Fig. 7c), 0.005 (Fig. 7e), and 2° (Fig. 7g) except for a few stripes in the low SNR region. These stripes may also be affected by the water tower nearby. Most of the CPPAR data points have consistent values throughout the PPI scans for all six radar variables, which is a promising feature. There are also no clear variations with scanning angle, and/or interference-like structures that could indicate

interferences/creeping wave effects.

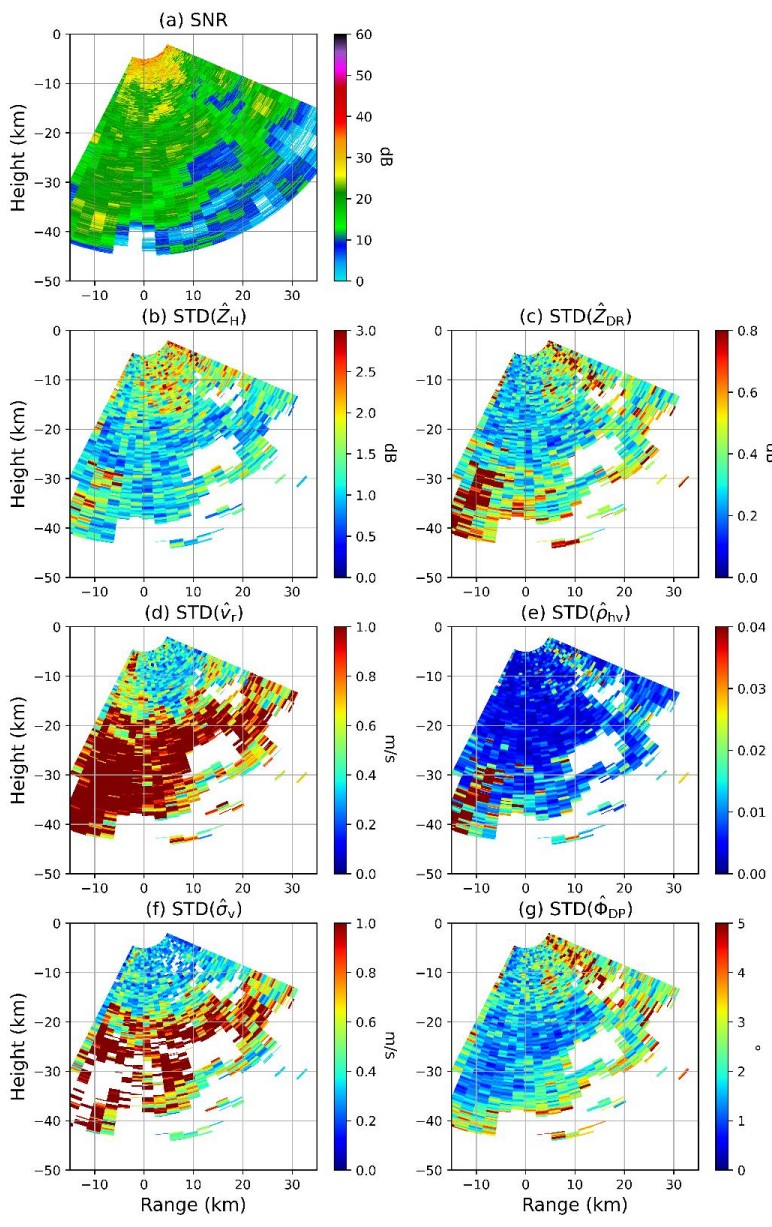

**Figure 7.** Same as Figure 6, except for CPPAR using spatial samples on 30 August 2019.

Initially presented in Table 3 of Li et al., (2021), the standard deviation estimates have been extended to cover
additional SNR ranges (Table 5). The results in Table 5 are very similar to those in Table 4a, showing a decreasing
trend in the standard deviation with increasing SNR. Furthermore, the standard deviation values for CPPAR are
comparable to those of Horus within the same SNR ranges. In general, Horus displays slightly smaller standard



deviations compared to CPPAR. This difference can be attributed to the use of oversampling with elevation for Horus. It should also be noted that CPPAR conducted electronic scans between full −45° and 45°, whereas Horus only −31.5° and 31.5°. For SNR values greater than 20 dB, the standard deviations observed in CPPAR are either less than or comparable to the theoretical values and are in compliance with the NOAA/NWS RFR for the examined case. Overall, the standard deviation values in Figure 6 and Figure 7 exhibit reasonable distributions, consistent with the results in Table 4 and Table 5. This indicates that the results are consistent across different analyses, providing a comprehensive assessment of the data quality for both radar systems.

|  | 0≤SNR | 5≤SNR<10 | 10≤SNR<15 | 15≤SNR<20 | 20≤SNR | Theory | NOAA/NWS RFR |
|---|---|---|---|---|---|---|---|
| $Z_H$ (dB) | 1.31 | 1.02 | 0.93 | 0.93 | 1.34 | 1.31 | 1.8 |
| $v_r$ (m/s) | 0.73 | 0.87 | 0.94 | 0.96 | 0.52 | 0.5 | 1.0 |
| $\sigma_v$ (m/s) | 0.54 | 0.79 | 0.74 | 0.76 | 0.36 | 0.34 | 1.0 |
| $Z_{DR}$ (dB) | 0.40 | 0.72 | 0.48 | 0.35 | 0.30 | 0.26 | 0.3 |
| $\rho_{hv}$ | 0.009 | 0.034 | 0.014 | 0.007 | 0.004 | 0.004 | 0.006 |
| $\Phi_{DP}$ (°) | 2.30 | 4.72 | 2.87 | 1.84 | 1.69 | 1.7 | 2.0 |

**Table 5.** Same as Table 4a, except the STDs are based on electronic scan of CPPAR.

### 4. Analysis on the off-broadside dilemma of planar configuration

The primary concern with the planar antenna design of 2D electronic PPARs lies in the weather data quality off broadside due to the scan-dependent beam properties of the PPPAR, unless an accurate beam-to-beam calibration is performed, which is difficult to do. While the planar configuration is relatively easy to implement and has been chosen by many fields for their applications (Brookner, 2008), including Horus, the PPPAR has inherent off-broadside problems that cause sensitivity loss in $Z_H$ and $Z_{DR}$ bias. It requires a larger antenna size, higher transmit power, complicated beamforming, and polarimetric calibration when the beam is off the principal plane or far away from the broadside. Previous literatures (e.g., Zhang et al., 2011, Golbon-Haghighi et al., 2021; Zhang et al., 2022) have warned of such fundamental challenges of electronically scanning PPPARs based on theoretical analysis and simulations. These limitations have been observed by the NSSL Advanced Technology Demonstrator (ATD) where data quality degradation occurs at wide scanning angles (Ivić et al., 2019). This study reveals a glimpse of such limitations based on the 13-panel observations from Horus. Horus only performs RHI scans in the vertical principal plane in its current development state, but the effect of off-broadside scanning can be inferred by comparing the quality of the polarimetric variables. As discussed in relation to Figure 6, the large area of reduced $\rho_{hv}$ could be caused by a combination of physical causes from the low intrinsic values in the melting layer, low SNR, or wide-angle steering off the broadside. The standard deviation of $Z_{DR}$, $\rho_{hv}$, and $\Phi_{DP}$ increased at the location of the reduced $\rho_{hv}$ as stripes, strengthening the possibility of an issue in electronic scanning off broadside.

To further investigate the potential cause of the large $\rho_{hv}$ reduction and stripes of higher standard deviations, the bias calculated from the KTLX comparison, and the standard deviations estimated from the spatial samples were averaged for each elevation angle. In order to isolate the problem, regions with SNR less than 20 dB, and KTLX $\rho_{hv}$ less than 0.95 were excluded from the analysis to minimize the effect of other causes such as ground clutter, melting level, and low SNR. Also, only rays with more than 300 valid values were considered for statistical significance. Note that CPPAR is not plotted as CPPAR does not have an angular dependence issue, and the data is insufficient for statistical significance after applying those filters. Figure 8 depicts the averaged bias for each ray with respect to the

broadside angle. For $Z_H$, there is a high bias in the lowest few elevations, probably from the remaining effect of ground clutter. Also, the reconstructed KTLX measurements in the lowest elevations are partially interpolated from the upper elevations increasing the potential for beam mismatch. Overall, the bias remains relatively stable around 3.5 dB. However, the polarimetric variables, $Z_{DR}$ and $\rho_{hv}$, reveal different error characteristics with respect to the broadside angle. While there are slight fluctuations, $Z_{DR}$ shows a negative trend, with much higher biases in the lower elevations

(i.e., 0° to −31.5°), and lower biases in the higher elevations (i.e., 0° to 31.5°) compared to the broadside. While the $Z_{DR}$ calibration factor can reduce biases in the broadside, the varying error characteristics (i.e., positive bias in the lower elevations and negative bias in the higher elevations) remain as the beams are steered away from the broadside. In fact, such biases start to be noticeable after ~20°, which is consistent with previous studies (e.g., Figs. 6 & 7 of Ivić, 2023; Fig.2 of Zrnic et al., 2011). Similar trends can be observed for $\rho_{hv}$, where large negative biases are evident from

~20°. The reduction in higher elevations is minor, but the decreasing trend can be seen with a reduction of ~ 0.002 compared to the broadside.

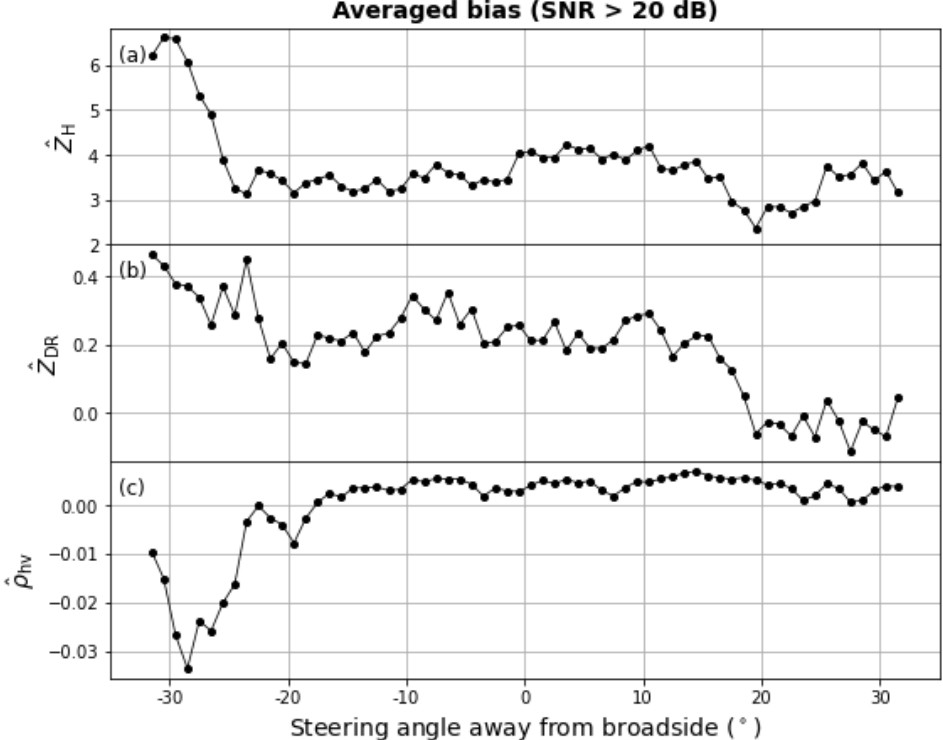

**Figure 8.** Plot of the averaged bias of $Z_H$, $Z_{DR}$, and $\rho_{hv}$ for each steering angle away from the broadside. Only SNR greater than 20 dB, and KTLX $\rho_{hv}$ greater than 0.95 were considered.

The averaged standard deviation plots also reveal similar features (Figure 9). All polarimetric variables depict a nearly parabolic shape with a minimum near the broadside (Figs. 9b, 9d, and 9f). The single polarization variables (Figs. 9a, 9c, and 9e) show random fluctuations for $Z_H$, and decreasing trend with elevation for $v_r$ and $\sigma_v$ as can be expected for severe storms. As the propagation path is ~4 times different for lower and upper elevations, the



magnitudes of the standard deviations between upper and lower elevations are not expected to be similar. For $Z_{DR}$, much larger standard deviation estimates up to 0.5 dB are noticeable even after removing low SNR and $\rho_{hv}$ regions, with a decreasing trend closer to the broadside (Fig. 9b). For the higher elevations, an increasing trend is noticeable away from the broadside. $\rho_{hv}$ shows a clearer parabolic shape, with values up to 0.023 at lower elevations and ~0.027 at higher elevation compared to less than 0.005 near the broadside (Fig. 9d). Standard deviation of $\Phi_{DP}$ also show

increasing trend away from the broadside, even after neglecting the high peak at the lowest elevation (Fig. 9f). As shown for the polarimetric biases in Figure 8, there is an increase in the error characteristics after electronically scanning away from the broadside. Thus, such a large $\rho_{hv}$ reduction in Horus measurements may be caused by issues associated with copolar beam mismatch and polarization purity loss at wide-angle steering off broadside, or interference from sidelobes. It can be expected that such performance degradation will be even worse when the beam

steers in a wide angle range from −45° to 45°.

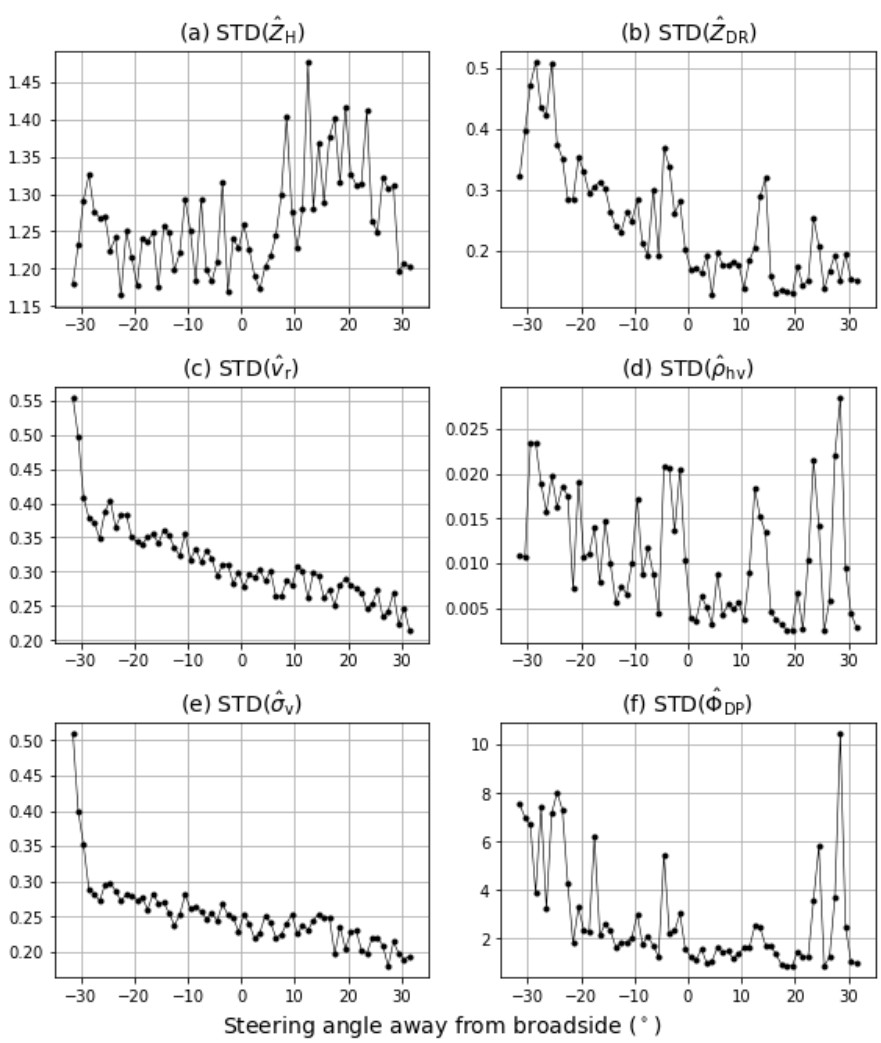

**Figure 9.** Same as Fig.8 except for averaged standard deviation of $Z_H$, $Z_{DR}$, $v_r$, $\rho_{hv}$, $\sigma_v$, and $\Phi_{DP}$.

These results are consistent with previous studies, which have found that polarimetric calibration for each element and direction of PPPAR is more tedious and difficult compared to the cylindrical configuration. This is because the active element patterns in PPPAR are different and difficult to characterize and form high-performance beams in all directions, while the active element/column patterns in CPPAR are all the same, yielding all formed beams the same with low sidelobes (e.g., Dorsey et al., 2022; Golbon-Haghighi et al., 2021; Logan et al., 2022; Zhang et al., 2011). Such limitations may reduce the usefulness of 2D PPPAR for accurate weather measurements, particularly as the antenna size increases and more elements are employed.



On the other hand, CPPAR provides all beam measurements in principal planes and with small angles from broadside, making it more immune to the degradation problems of electronic steering at wide angles. Nevertheless, the development of CPPAR in full-scale of the WSR-88D has been stopped, even though the main issues are solved with satisfactory results. For a better understanding of such limitations of PPPAR and other possible deficiencies of CPPAR, observations of the same case with various scanning strategies of the 2D electronic PPARs should be conducted in the future.

## 5. Assessing advantages and disadvantages of planar and cylindrical configurations

The two of the most promising configurations, planar and cylindrical, of 2D PPARs have received considerable attention over the past decade to explore future applications. The timeline and goals depicted in Fig.10 are based on Fig. 1 of NSSL MPAR report (2014), with a few modifications reflected to align with the actual timeline. Initially, the NSSL, (2014) report planned to analyze both configurations (i.e., 4 faced PPPAR and CPPAR) without any moving parts to select the most optimal design for accurate meteorological measurements and potential multifunctionality. Both configurations have been developed as in the 10-panel demonstrator (TPD) and CPPAR-I, and further efforts have been planned and invested for ATD and CPPAR-II. However, the plan changed over the years, and these 2D PPARs currently focus on weather measurements in providing efficient and high-quality PRD in all directions, especially at angles far off broadside (NSSL, 2017, 2023). Another notable deviation from the original plan concerns the two 2D PPPARs, Horus and ATD, while CPPAR development has stopped since 2020. As a rough estimation, much more investment was allocated to each of the planar designs over CPPAR, as evidenced by the much narrower beamwidth and transmit power of Horus compared to CPPAR, and the number of T/R modules. However, as shown in this study, the quality of weather measurements from these PPPARs at wide angles are questionable.

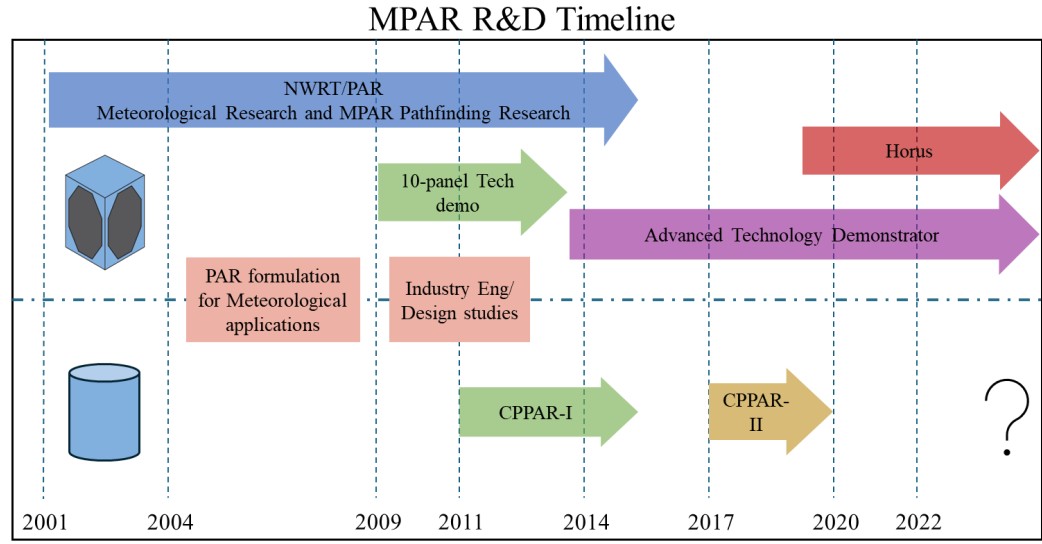

**Figure 10.** Reconstructed MPAR research and development timeline based on Fig.1 of NSSL (2014).



The discussions on the optimal configuration (e.g., planar vs. cylindrical), and/or limitations of 2D PPPAR have been well documented and presented over the years (Doviak et al., 2011; Ivić, 2023; Karimkashi and Zhang, 2015; Lei et al., 2013, 2015; NSSL, 2014, 2020a, 2020b, 2021; Zhang et al., 2009, 2011; Zrnic et al., 2011). The advantages and disadvantages of each configuration, based on those previous studies and the quantitative error analysis of weather measurements conducted in this study, are organized in Table 6. As analyzed in several previous studies and confirmed in section 4, 2D PPPARs have inherent problems with i) beam broadening, ii) sensitivity loss, iii) loss of polarization purity, and iv) higher risk of beam mismatch when steering off-broadside. These problems arise because the patterns of the elements embedded in the PPPAR vary from one to another and their patterns are not the same, making it difficult to achieve high performance PPPAR beams (e.g., polarization purity, matched dual-polar beams, low side lobes, etc.). These issues can be further assessed by beam-to-beam pattern characterization and weather measurements with multi-beams. Some of these problems can be corrected by calibration, such as phase coding or appropriate antenna tilt (Ivić, 2022, 2023), but the others can only be avoided/resolved through configuration considerations, antenna design, and optimal beamforming. While this study's analysis emphasized the off-broadside problem, as Horus only conducted RHI scans and CPPAR PPI, scanning off the principal plane can present an even greater challenge. Planar design is often chosen for many other applications because most do not require wide-angle scans or only require qualitative data, such as aircraft detection. However, this is not the case for meteorological applications, which require high-quality quantitative polarimetric measurements. Specifically, $\rho_{hv}$ requires an error of less than ~0.006.

| CPPAR | PPPAR (Horus, ATD) |
|---|---|
| + Azimuthally scan invariant beam | − Scan variant beam and scan dependent biases |
| + Always scanning in principle plane | − Loss of sensitivity when scanning off principal plane |
| + Small angles from broadside | − Loss of polarimetric purity when scanning off-broadside or |
| + Polarimetric purity and easy polarimetric calibration | principal plane |
| | − Difficulties in polarimetric calibration |
| + Interference of creeping wave not shown in CPPAR data | − Surface wave can exist |
| o New design and concept for the community and the industry | + Mature design for the community and industry |
| | + Flexible beam steering from isolated face |

**Table 6.** The table lists the advantages (+), disadvantages (-), and neutral (○) of each 2D PPAR configuration.

CPPAR was chosen for 2D PPAR to mitigate issues such as beam broadening, sensitivity loss, and polarization purity. CPPAR always scans in the principal plane in azimuth with small angles in elevation, using the same physical principles as 1D PPPAR and producing polarimetric data of comparable quality (as demonstrated in Li et al., 2021). In addition, high-performance radiating elements have been designed, ideal arrangements were proposed, and beams were optimally formed for CPPAR so that the mentioned creeping wave effect is not an issue (Golbon-Haghighi et al., 2021; Mirmozafari et al., 2017, 2019; Saeidi-Manesh et al., 2017). CPPAR has produced high-quality polarimetric weather data, which has been quantitatively evaluated (Li et al., 2021 and this study). Although there were concerns regarding limitations of CPPAR, such as interference or creeping waves, the analysis and demonstration of CPPAR measurements do not show these issues due to their minimal effect. This is because CPPAR beams were optimally


formed from the active (embedded) element/column patterns using the multi-objective optimization so that the dual-pol beams are well-matched and sidelobes are low. Although the creeping/surface effects appear in the active element patterns as ripples, the CPPAR beams formed from the active patterns have already taken these effects into account and therefore are almost the same for all the beams with high performance (see Figs. 4&5 of Zhang, 2022). The beam characteristics can be further improved using dipole antennas and/or larger sizes (Mirmozafari et al., 2019; Golbon-Haghighi et al., 2021). CPPAR is relatively new to the community and industry. However, its feasibility is evidenced by CPPAR data and this study, making it a promising research project aimed at advancing future weather measurements.

The cost-performance trade-offs among the current operational dish, 1D, and 2D PPARs are summarized in Table 7, taking into account previous studies and the results of this study. Dish-based current operational radars provide relatively cheap and high-quality weather measurements, but have much slower update speed. 1D PPPAR and CPPAR share the same physical principles (i.e., electronic scanning in the principal plane with small steering angles), with differences in cost and update speed/flexibility. It is generally agreed that 1D PPPAR is feasible and cost-effective for weather observation, and 1D X-band PPPARs have already been deployed for operational use in the USA, Japan, and China. Considering that a 4-faced 2D PPPAR is more expensive than CPPAR and 1D PPPAR, and given the known shortcomings and beamforming&calibration difficulties in providing high-quality data in all directions, a rigorous quantification of its performance is necessary before pursuing 2D PPPAR for weather measurement.

|  | Dish | 1D PPPAR | 2D PPPAR | CPPAR (2D) |
|---|---|---|---|---|
| **Cost** | Low | Medium | High | High |
| **Data quality** | High | High | Low | High |
| **Update speed** | Low | Medium/High | High | High |
| **Calibration/maintenance** | Easy | Easy/Moderate | Difficult | Moderate |
| **Potential** | Low | High (short-term) | Low | High (long-term) |

**Table 7.** The advantageous and disadvantageous properties of dish radar, 1D PPPAR, 2D PPPAR, and CPPAR for cost, data quality, update speed, calibration/maintenance methods, and potential criteria.

## 6. Summary

This study presents the first quantitative error analysis and comparison of 2D PPPAR and CPPAR data. It is shown that both PPPAR and CPPAR can provide accurate polarimetric weather measurements when their beams are close to the broadside. The PPPAR performance degrades as the beam steers away from the broadside. It is worth noting that there are several limitations of the study, including differences in range resolution between dish-based radars and PPPARs, and problems with the absolute calibration of the polarimetric variables of the KTLX radar. Despite these limitations, and the unavailability of observations of the same event from co-located radars with identical scanning strategies, the study provides valuable insights by comparing weather measurements from two promising PPPAR configurations: planar and cylindrical. The standard deviations and mean biases of the Horus and CPPAR measurements were calculated and compared with those of an operational WSR-88D radar. The standard deviations for both PPPARs agree well with theoretical expectations and are within the NOAA/NWS RFR for the cases studied. The standard deviation from spatial samples shows smaller values compared to that from temporal samples, which



may be attributed to post processing filters used in Horus. Bias calculations relative to KTLX show 3.99 dB for $Z_H$, 0.23 dB for $Z_{DR}$, and −0.002 for $\rho_{hv}$ for Horus, and −1.29 dB for $Z_H$, 0.04 dB for $Z_{DR}$, and 0.009 for $\rho_{hv}$ for CPPAR.

The study highlights the inherent limitations and challenges of polarimetric calibration for 2D PPPARs, particularly due to loss of polarization purity and mismatch of dual-polarimetric beams when steering away from the broadside or principal planes. It should be noted that 2D PPPARs will need to steer off the principal planes and perform even wider angle scans. This study only demonstrated the effect of limited off-broadside scans, but has already revealed potential deficiencies. Accurate weather measurements with PPARs require an understanding of scanning loss issues and system performance at each beam steering direction in order to apply appropriate calibrations. Dual-scan comparisons and multi-pattern measurements have been effective in analyzing this aspect.

To avoid the inherent limitations for 2D PPPAR, the cylindrical configuration is another option, which has the greatest potential to enhance current operational radars by providing high-quality polarimetric data and rapid data updates. These advances are likely to improve weather forecasting and the understanding of rapidly changing weather phenomena, particularly severe storms.

**Competing interests**

The contact author has declared that none of the authors has any competing interests

**Acknowledgements**

This work was partially supported by the National Oceanic and Atmospheric Administration (NOAA) under Grant NA16OAR4320115 and the National Science Foundation with the grant of AGS-2136161. The views expressed are those of the authors and do not necessarily represent the official policy or position of the U.S. Government. The
authors would like to thank Dr. Berrien Moore III and the Horus engineering team for their assistance in providing the data.

**Data Availability Statement**

CPPAR and Horus data used in this study are available upon request at data@arrc.ou.edu. The WSR88D level-II data were obtained from the National Centers for Environmental Information (www.ncei.noaa.gov).

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
