# Peer review of "Quantitative Error Analysis on Polarimetric Phased Array Radar Weather Measurements to Reveal Radar Performance and Configuration Potential"

_Atmospheric Measurement Techniques, 2024_

## Author Response (AR1)

Reviewer #1:

This document discusses a study comparing two types of polarimetric phased array radars (PPARs) developed by the Advanced Radar Research Center (ARRC) at the University of Oklahoma: a cylindrical PPAR (CPPAR) and a planar PPAR (PPPAR) named Horus. The study aims to evaluate these radars' performance and error characteristics in weather measurements, using data from the operational KTLX Weather Surveillance Radar-1988 Doppler (WSR-88D) as a reference.

Reply: Thank you for your insightful comments. Below are our point-by-point responses and the changes made to the manuscript, highlighted in blue:

Comment 1: What specific design improvements have been made to address CPPAR's challenges, and how do they contribute to its performance?

Reply: The earlier CPPAR system, CPPAR-I, suffered from electronic instability due to inexpensive digital transceivers used and dual-polarization beam mismatch as well as clutter contamination. These problems were exacerbated by the fact that the system was mounted on a trailer on the ground. As a result, $\rho_{hv}$ was biased low and high-quality polarimetric radar data (PRD) could not be obtained for weather observations (Byrd et al. 2017; Zhang 2022). In CPPAR-II, several improvements were made, including: (i) redesigning the column antenna with matched dual-polarization patterns such that the a pointing angle mismatch is less than ±0.2° (<4% of the beamwidth) (Saeidi-Manesh et al. 2017), (ii) switching from digital to analog beamforming to improve system stability, and (iii) optimally forming the CPPAR beams using multi-objective optimization techniques (e.g., Karimkashi and Zhang 2015; Golbon-Haghighi et al. 2018; Li et al. 2021; Zhang 2022). Along with the inherent advantages of azimuthally scan-invariant beams and polarization purity for CPPAR, the nearly identical high-performance beams allow for simple calibration and accurate polarimetric weather measurements. The data used in this manuscript were collected after the system was modified and deployed on the RIL rooftop (see Figure 1b) with ground clutter effects substantially reduced. The following will be included in the revised manuscript for further information.

Lines 481–486: Many of the problems and lessons learned from CPPAR-I have been addressed and resolved over the years. Several improvements include: (i) redesigning the column antenna with matched dual-polarization patterns (Saeidi-Manesh et al. 2017), (ii) switching from digital to analog beamforming to improve system stability, and (iii) optimized beamforming to achieve the nearly identical high-performance beams using multi-objective optimization techniques (e.g., Karimkashi and Zhang 2015; Golbon-Haghighi et al. 2018; Li et al. 2021; Zhang 2022)

Comment 2: Lower overall CPPAR sensitivity may limit the data's accuracy and comprehensiveness. How much of the lower Signal-to-noise ratio (SNR) introduces biases in the analysis?

Reply: It is true that the lower sensitivity of CPPAR limits the measurement coverage to regions of higher reflectivity. However, the lower SNR is not expected to substantially introduce biases in the error analysis, because we categorize the errors by SNR levels. Additionally, the standard deviation (STD) of the errors depends on the spectrum width and the copolar correlation coefficient for the same dwell time. The bias and STD of CPPAR for different SNR ranges, as shown in Tables 3 and 5, are in accord with theory. The bias remains relatively stable across SNR thresholds, while the STD decreases with increasing SNR due to the factors mentioned above. The following will be included in the revised manuscript for clarification.

Lines 139–144: As expected from the parameters of the three radars, the two 2D PPARs have a much lower sensitivity of 25 dBZ for CPPAR, and 10 dBZ for Horus, compared to −10 dBZ for the operational KTLX at 45 km away from the radar. The lower sensitivity is not expected to substantially introduce biases in the error analysis. The errors are categorized by SNR levels, and the standard deviation (STD) of the errors depends on the spectrum width and the copolar correlation coefficient for the same dwell time.

Comment 3: How can concerns be mitigated that the planar antenna design may not effectively mitigate the weather data quality off broadside due to the scan-dependent beam properties of the PPPAR?

Reply: Given that performance degradation as the beam steers away from the broadside is inherent and fundamental, mitigation is limited. Possible mitigation approaches include: (i) avoiding steering to large angles away from the broadside, (ii) accurately characterizing embedded element patterns and optimally forming beams at each steering direction, (iii) performing beam-to-beam calibration for PPPAR, and (iv) using a 1D electronic scan PPPAR, where the main beam always remains in the principal plane, minimizing cross-coupling. A few other potential approaches have been documented, such as antenna tilt (Ivic 2023) or phase coding (Ivic 2022). However, as noted in these studies, compensating for the reduced $\rho_{hv}$ caused by polarization purity loss and scanning off broadside remains very challenging, which is supported by the results in this manuscript. It is also important to note that the Horus measurements in this study were only performed from −33° to +33°, and further degradation is expected when scanning from −45° to +45°, as required for the 4-face PPPAR system. The following will be included in the revised manuscript.

Lines 515–519: For example, the performance degradation away from the broadside can only be mitigated by (i) avoiding steering at large angles away from the broadside, (ii) accurately characterizing embedded element patterns and optimally forming beams at each steering direction, (iii) performing beam-to-beam calibration for PPPAR, and (iv) using a 1D electronic scan PPPAR, where the main beam always remains in the principal plane, minimizing cross-coupling.

Comment 4: Comment on how the study's exclusion of certain data, such as regions with SNR less than 20 dB and KTLX ρhv less than 0.95, may impact the overall conclusions drawn about the limitations of electronically scanning PPPARs.

Reply: Including data with SNR < 20 dB and $\rho_{hv}$ < 0.95 would increase the errors in polarimetric measurements, leading to further degradation in electronically scanning PPPARs. In this study, as the first analysis of its kind, we focus on the high SNR case for simplicity and demonstration purposes. In addition, Horus performed RHI scans, and the $\rho_{hv}$ threshold was applied to eliminate the influence of melting layers, allowing the focus to remain on the analysis of error caused by the system and sampling. The following will be included in the revised manuscript for clarification.

Lines 421–423: The removal of SNR and $\rho_{hv}$ threshold would increase the errors in polarimetric measurements, leading to further degradation in electronically scanning PPPARs. In this study, as the first analysis of its kind, we focus on the high SNR case for simplicity and demonstration purposes.

Comment 5: How do you generalize that the study's findings may not fully represent the challenges associated with electronically scanning PPPARs, given the exclusion of CPPAR data and limited statistical significance after applying certain filters?

Reply: We acknowledge that the analysis and results presented in this paper may not fully capture all the challenges of 2D electronically scanning (E-Scan) PPPARs due to the limited data. However, this is the best we could achieve with the data and resources available. Since the analysis follows established physical principles, the results could be refined with additional data, but the underlying fundamentals should remain unchanged (e.g., Zhang et al. 2011; Ivic 2022, 2023). The following will be included in the revised manuscript for clarification.

Lines 575–576: In addition, the analysis and results presented in this paper may not fully capture all the challenges of 2D electronically scanning (E-Scan) PPPARs due to the limited data.

References

Byrd, A., C. Fulton, R. Palmer, S. Islam, D. Zrnic, R. Doviak, R. Zhang and G. Zhang, "First Weather Observations with a Cylindrical Polarimetric Phased Array Radar," OU/ARRC Internal Technical, Report, 2017.

Karimkashi, S. and Zhang, G.: Optimizing radiation patterns of a cylindrical polarimetric phased-array radar for multimissions, IEEE Transactions on Geoscience and Remote Sensing, 53(5), 2810–2818, doi:10.1109/tgrs.2014.2365362, 2015.

Saeidi-Manesh, H., Mirmozafari, M. and Zhang, G.: Low cross-polarisation high-isolation frequency scanning aperture coupled microstrip patch antenna array with matched dual-polarisation radiation patterns, Electronics Letters, 53(14), 901–902, doi:10.1049/el.2017.1282, 2017.

Golbon-Haghighi, M.-H., Mirmozafari, M., Saeidi-Manesh, H. and Zhang, G.: Design of a cylindrical crossed dipole phased array antenna for weather surveillance radars, IEEE Open Journal of Antennas and Propagation, 2, 402–411, doi:10.1109/ojap.2021.3059471, 2021.

Li, Z., Zhang, G., Golbon-Haghighi, M.-H., Saeidi-Manesh, H., Herndon, M. and Pan, H.: Initial observations with electronic and mechanical scans using a cylindrical polarimetric phased array radar, IEEE Geoscience and Remote Sensing Letters, 18(2), 271–275, doi:10.1109/lgrs.2020.2971471, 2021.

Zhang, G.: Cylindrical polarimetric phased array radar for weather observations: A Review, 2022 IEEE Radar Conference (RadarConf22), doi:10.1109/radarconf2248738.2022.9764328, 2022.

Zhang, G., Doviak, R. J., Zrnić, D. S., Palmer, R., Lei, L. and Al-Rashid, Y.: Polarimetric phased-array radar for weather measurement: A planar or cylindrical configuration?, Journal of Atmospheric and Oceanic Technology, 28(1), 63–73, doi:10.1175/2010jtecha1470.1, 2011.

Ivic, I. R.: Quantification of polarimetric par effects on weather observables in the phase coded STSR Mode, IEEE Transactions on Geoscience and Remote Sensing, 60, 1–22, doi:10.1109/tgrs.2022.3146358, 2022.

Ivić, I. R.: Cross-coupling mitigation in polarimetric par via antenna tilt, Journal of Atmospheric and Oceanic Technology, 40(5), 587–604, doi:10.1175/jtech-d-22-0059.1, 2023.

Reviewer #2:

This study compares polarimetric radar data quality between two phased-array antenna configurations – a planar phased array (PPAR) and a cylindrical commutating array (CPPAR) using a nearby NEXRAD radar (KTLX) as a reference. Biases and standard deviations are examined between the two configurations through case studies of opportunity by pairs of the radars: PPAR vs KTLX in Oct 2023 and CPPAR vs KTLX in August 2019. The PPAR comparison uses RHI scans at broadside azimuth (i.e. along a principle plane). The CPPAR comparison uses PPI scans. The radial paths to the target areas differ for the three radars, their sampling volumes differ considerably, as do the signal-to-noise ratios. Reasonable attempts are made to account for these differences.

Thank you for your insightful comments and questions. Below are our point-by-point responses and the changes made to the manuscript, highlighted in blue:

Comment 1: Based on statistics derived from the case studies It is argued that the polarimetric quality is superior in the CPPAR configuration owing to always scanning in a principle plane. While I believe this may be true, it is not clear that this is proven in this study, as the PPAR observations were also made in the principle plane. What is observed is that PPAR comparisons with KTLX were made at an off broadside direction (in elevation). Please clarify how far off-broadside.

Reply: The CPPAR configuration is superior because it scans in the principal plane with azimuthally invariant beams and operates within close steering angles in elevation (with a maximum elevation angle of 20 degrees for the WSR-88D). This means it doesn't have to scan far from the broadside. For the data provided, Horus only performed scans from −31.5° to 31.5° in elevation, without scanning in azimuth. However, as a 2D PPPAR (one of four faces), it is expected to eventually electronically scan from −45° to +45° degrees in azimuth, which would make PPPAR performance even worse at the large steering angles. Additional clarifications in the summary section of the manuscript will be changed as follows.

Lines 585–589: The study highlights the inherent limitations and challenges of polarimetric calibration for 2D PPPARs, particularly due to the scanning loss, the loss of polarization purity, the mismatch of dual-polarimetric beams and the difficulty in controlling the sidelobes when steering away from the broadside. It should be noted that 2D PPPARs will need to steer away from the principal planes and perform even wider angle scans. This study has only demonstrated the effect of limited off-broadside scans, but has already revealed potential deficiencies.

Comment 2: The PPAR appears to have been aimed rather high, which is not how one might envision practical measurements to be made. Radar polarimetry is predicated on near-horizontal propagation, so basing claims on measurements made with the antenna oriented at a high angle may not be representative of typical performance. Please elaborate on this.

Reply: Yes, it is not the optimal condition since weather radars typically only scan up to ~20° in elevation. However, this is one of the earlier measurements from the PPARs, and is only conducting 1D scans in elevation. It is analogous to scanning in azimuth and demonstrates the potential issues with 2D PPPARs scanning far off broadside on the principal plane. The following sentence will be added to the manuscript.

Lines 115–118: While it is true that most weather radars scan up to ~20° for weather applications, Horus in its current state only performs a 1D electronic scan along the elevation between −31.5° to 31.5°, and should reveal similar problems as scanning in azimuth in the similar weather condition.

Comment 3: Line 490: It is argued that PPARs are harder to calibrate because the elements vary from one to another, while the elements in CPPAR are all the same. This seems disingenuous. The elements in both configurations are all designed to be the same, but variations inevitably occur due to electrical or mechanical tolerances that result in element-to-element errors. It is hard to accept that these kinds of errors do not occur in the CPPAR. Please explain further.

Reply: It is true that all elements in both configurations are designed and constructed in the same way, resulting in identical isolated element patterns when the elements radiate individually. However, in an array antenna environment, the elements radiate in the presence of surrounding elements, leading to active (or embedded) element patterns that differ from the isolated patterns due to mutual interactions among the radiating elements. Furthermore, these active/embedded element patterns vary depending on the location of the element within the array due to differences in their electromagnetic environments. For example, a central element typically exhibits a symmetric pattern if designed as such, whereas elements at the sides or corners may have asymmetric patterns due to edge or corner effects. This variation occurs in PPPAR but not in CPPAR which maintains symmetry—ensuring that all columns are in the same electromagnetic environment and therefore have the same embedded element patterns (see Fig. 4 of Zhang 2022). As a result, the same properties can be applied to all columns, which simplifies the beamforming and calibration in CPPAR. Further details will be given in the revised manuscript.

Lines 505–511: Although all elements are designed identically in both configurations, resulting in the same isolated element patterns, the array antenna configuration causes active/embedded element patterns being different due to the presence of surrounding elements. These active element patterns vary depending on the location of the element within the array (i.e., its electromagnetic environment). For example, the central element may have a symmetric pattern if appropriately designed, while elements at the sides or corners may have asymmetric patterns due to edge or corner effects. This variation occurs in PPPAR but not in CPPAR which maintains symmetry—ensuring that all columns have the same electromagnetic environment and therefore the same embedded column patterns.

Comment 4: In figures 8 and 9, please clarify the elevation angles that correspond to the steering angles. I assume that -30 is closer to horizontal and +30 is closer to vertical. To what extent may some of these be attributable to ground clutter contamination?

Reply: The antenna patterns of the whole array at these steering angles are not accessible to the authors of this manuscript.

Based on the beamwidth, ground clutter contamination is expected to affect only up to three to four lowest elevation angles in the main beam, as the elevation beamwidth is 3.3° and measurements were sampled every 1°. However, it is also important to note that oversampling may have affected the standard deviation of the Horus measurements. Although the error statistics show a slight decrease, no significant decrease is observed, as shown below. In addition, the $\rho_{hv}$ filter was applied to minimize the effect of ground clutter and melting layer.

The captions for Figures 8 and 9 will be updated as follows.

| Standard deviation | All elevations | All elevations with $\rho_{hv}$ filter | Removed four lowest elevations |
|---|---|---|---|
| $Z_H$ (dB) | 1.69 | 1.68 | 1.71 |
| $v_r$ (m/s) | 0.44 | 0.44 | 0.43 |
| $\sigma_v$ (m/s) | 0.35 | 0.35 | 0.34 |
| $Z_{DR}$ (dB) | 0.25 | 0.24 | 0.23 |
| $\rho_{hv}$ | 0.005 | 0.005 | 0.004 |
| $\Phi_{DP}$ (°) | 1.72 | 1.68 | 1.55 |

| Bias | All elevations | Removed four lowest elevations |
|---|---|---|
| $Z_H$ (dB) | 3.99 (4.33) | 3.47 (3.83) |
| $Z_{DR}$ (dB) | 0.23 (0.29) | 0.25 (0.27) |
| $\rho_{hv}$ | −0.002 (0.003) | −0.001 (0.003) |

Figure 8. Plot of the averaged bias of $Z_H$, $Z_{DR}$, and $\rho_{hv}$ for each steering angle away from the broadside. Only SNR greater than 20 dB, and KTLX $\rho_{hv}$ greater than 0.95 were considered. Note that the angles range from −31.5° to 31.5°.

Comment 5: Table 6 seems incomplete. The issues of creeping or surface waves potentially exist in any phased array design. If designed properly, they should not occur, so should not be included in this table. While the CPPAR does have a polarimetric quality advantage, there are other disadvantages associated with scanning limitations (i.e. commutation) and aperture efficiency that do not appear to be indicated.

Reply: Yes, creeping or surface waves can potentially occur in any phased array design. Factors such as mutual coupling between elements, cross-coupling between polarizations and their effects

on the formed beam characteristics should be considered in this context. While mutual coupling can be used for calibration (e.g., Fulton et al. 2022), it can also cause the loss of polarization purity and difficulty in controlling sidelobes. In addition, there is still confusion and misunderstanding within the weather radar community about the risk and interferences caused by surface/creeping waves in CPPAR (e.g., NSSL 2014, 2023). Actually, this issue has been addressed such that the beams have been formed with high polarization purity and very low sidelobes, and reported (e.g., NSSL 2020; Golbon et al. 2021) as well as demonstrated with CPPAR weather measurements (Li et al. 2021). As you correctly stated, it should be noted and clarified to the community that surface waves can also exist for planar configurations and is more difficult to mitigate because the surface wave effects vary from element to element.

Aperture efficiency has also been explored in previous studies, which indicate that the same number of elements is required for both cylindrical and planar configurations to achieve the same pencil beam (Zhang et al. 2011). In fact, power efficiency is generally better for cylindrical configurations because it forms beams in the principal plane and close to the broadside, avoiding significant scanning loss at wide angles that PPPAR has to steer.

The limitations of commutation scanning are now included in the table, although they may not be a big issue for weather applications. The manuscript will be clarified and elaborated as follows.

| CPPAR | PPPAR (Horus, ATD) |
|---|---|
| + Azimuthally scan invariant beam | − Scan variant beam and scan dependent biases |
| + Always scanning in principle plane | − Loss of sensitivity when scanning off principal plane |
| + Small angles from broadside | − Loss of polarimetric purity when scanning off-broadside or principal plane |
| + Polarimetric purity and easy polarimetric calibration | − Difficulties in polarimetric calibration |
| o New design and concept for the community and the industry | + Mature design for the community and industry |
| | + Flexible beam steering from isolated face |

Lines 532–536: There is still confusion and misunderstanding about interferences caused by surface/creeping waves in CPPAR within the meteorological community (e.g., NSSL 2014, 2023), but surface waves can exist in any configuration and should be considered and minimized, and it is easier to control the surface wave effects in CPPAR than in PPPAR (e.g., Mirmozafari et al. 2017 & 2019; Saeidi-Manesh et al. 2017; Golbon et al. 2021).

Lines 550–551: CPPAR has limitations in commutating scans and can have limited freedom compared to that of the planar, but this is not significantly relevant in the case of weather applications.

Comment 6: Similarly, Table 7 seems to be not data but the authors' opinions. Why does a dish radar have low potential? What do the authors man by potential?

Reply: Yes, Table 7 is not solely based on data, but summarizes the overall content of the manuscript. The authors believe that the table has been prepared in an objective manner. The "potential" rating includes all aspects of the criteria listed above in Table 7. While dish radars can provide high-quality weather measurements, the mechanical scanning in both azimuth and elevation limits their ability to improve update speed in the future. As you suggested, we now modified the dish antenna to have high potential for short term. 2D PPPAR was considered to have low potential due to their difficulty in providing high quality polarimetric data and high cost. The following sentences will be added to the manuscript for clarification.

Lines 554–558: The cost-performance trade-offs among the current operational dish radars, 1D, and 2D PPARs are summarized in Table 7, taking into account previous studies and the results of this research. The potential of each system is assessed based on a combination of criteria including cost, data quality, update speed, and calibration. Dish-based operational radars provide relatively low-cost and high-quality weather measurements; however, they have a much slower update speed, which limits their potential as the meteorological community is seeking faster update radars for the future.

|  | Dish | 1D PPPAR | 2D PPPAR | CPPAR (2D) |
|---|---|---|---|---|
| Cost | Low | Medium | High | High |
| Data quality | High | High | Low | High |
| Update speed | Low | Medium/High | High | High |
| Calibration/maintenance | Easy | Easy/Moderate | Difficult | Moderate |
| Potential | High (short-term) | High (mid-term) | Low | High (long-term) |

**References**

Golbon-Haghighi, M.-H., Mirmozafari, M., Saeidi-Manesh, H. and Zhang, G.: Design of a cylindrical crossed dipole phased array antenna for weather surveillance radars, IEEE Open Journal of Antennas and Propagation, 2, 402–411, doi:10.1109/ojap.2021.3059471, 2021.

NSSL Publications: Multi-function Phased Array Radar and Cylindrical Polarized Phased Array Radar Report to Congress: https://www.nssl.noaa.gov/publications/par_reports/ FY14_MPAR_CPPAR_Congressional_Report.pdf (Accessed 6 June 2024), 2014.

NSSL Publications: Feasibility and Capability of a rotating Phased Array Radar: https://www.nssl.noaa.gov/publications/par_reports/ Feasibility and Capability of a rotating Phased Array Report to Congress (Accessed 6 June 2024), 2023.

NSSL Publications: Spectrum Efficient National Surveillance Radar (SENSR): OAR Feasibility Study Final Report: https://www.nssl.noaa.gov/publications/par_reports/SENSR Final Report.pdf (Accessed 6 June 2024), 2020.

Mirmozafari, M., Zhang, G., Fulton, C. and Doviak, R. J.: Dual-polarization antennas with high isolation and polarization purity: A review and comparison of cross-coupling mechanisms, IEEE Antennas and Propagation Magazine, 61(1), 50–63, doi:10.1109/map.2018.2883032, 2019.

Mirmozafari, M., Zhang, G., Saeedi, S. and Doviak, R. J.: A dual linear polarization highly isolated crossed dipole antenna for MPAR application, IEEE Antennas and Wireless Propagation Letters, 16, 1879–1882, doi:10.1109/lawp.2017.2684538, 2017.

Saeidi-Manesh, H., Karimkashi, S., Zhang, G. and Doviak, R. J.: High-isolation low cross-polarization phased-array antenna for MPAR application, Radio Science, 52(12), 1544–1557, doi:10.1002/2017rs006304, 2017.